# Study on Mechanical Properties and Microstructure of Basalt Fiber-Modified Red Clay

**Yu Song** [1,2], **Yukun Geng** [1,2], **Shuaishuai Dong** [1,2], **Song Ding** [1,2], **Keyu Xu** [1,2], **Rongtao Yan** [1,2] and **Fengtao Liu** [1,2,*]

1   School of Civil and Architectural Engineering, Guilin University of Technology, Guilin 541004, China
2   Guangxi Key Laboratory in Geotechnical Mechanics and Engineering, Guilin 541004, China
*   Correspondence: celiuft@glut.edu.cn

**Abstract:** The effects of basalt fiber incorporation on the mechanical properties of red clay soils were investigated. Through the direct shear test, unconfined compressive strength test, and microstructure test, the shear strength curves and stress–strain curves of basalt fiber-modified red clay soils were obtained under different basalt fiber incorporation rates and different soil dry density conditions. The results showed that: (1) the shear strength and compressive strength of the soil were significantly increased after the incorporation of basalt fiber; (2) the strength increase was greatest at 0.3% of basalt fiber incorporation, which was the optimum incorporation level; (3) the damage form of the soil changed, and the red clay soil incorporated with basalt fiber changed from brittle damage to ductile damage; and (4) the microscopic electron microscope pictures showed that, at the appropriate amount of fiber incorporation conditions, the fiber bond with the soil particles and form a fiber-soil column. When subjected to external forces, the discrete fiber-soil columns interact with each other to form an approximate three-dimensional fiber-soil network, which acts to restrain the displacement and deformation of the soil particles, which is the main reason for the improved mechanical properties of the improved soil. The experimental research on the improvement of red clay soil with basalt fiber can provide a theoretical basis for engineering practice and help provide an environmentally friendly and efficient method of road base treatment in engineering.

**Keywords:** basalt fiber; red clay; mechanical properties; incorporation rate; microstructure





## 1. Introduction

Red clay soil is a soil with unique characteristics since it was formed in a region with high humidity, high temperature, and heavy precipitation. Due to its high water-absorption capacity, low bearing capacity, and substantial compression deformation, it may have negative impacts when utilized as foundation soil for filling the road base. In some regions of southern China, red clay is employed in the building as a fill soil. Therefore, it is essential to enhance its mechanical properties.

At present, calcium oxide, fly ash [1], volcanic ash [2], cement, and other materials are usually used as soil-curing agents. Research has found that fibers have good qualities and can interact with soil particles to boost the overall strength when mixed into the soil. The application of fibers in soil reinforcement in current engineering increases the strength of the soil, prevents the formation of tensile cracks, inhibits swelling soils that tend to swell, increases hydraulic conductivity and liquefaction strength, and reduces soil brittleness. Moreover, among numerous fibers, basalt fiber is an eco-friendly and pollution-free high-tech fiber.

The main manifestations are in the production process. First, basalt fibers do not produce harmful substances such as boron and other alkali metal oxides during the melting process. It is also inert, non-combustible, and non-explosive and does not produce toxic substances in contact with air and water, overcoming the disadvantages of traditional

glass fiber materials that consume a lot of energy and cause environmental pollution in the manufacturing process. Secondly, basalt fibers are ecologically friendly as they can degrade into soil parent material under natural conditions after disposal [3]. Basalt fiber has superior mechanical properties, electrical properties, wave permeability, non-conductivity, sound, and heat insulation properties compared to other extensively used fibers [4]. If the modified soil needs to be restored to its original state, the basalt fibers can be degraded by microorganisms. Early basalt fibers were more useful in the field of civil engineering to improve the performance of concrete. It was found that, by mixing in a small volume of basalt fiber, the compressive strength [5], toughness, and crack resistance of the concrete [6–10] could be improved. To realize the engineering application of basalt fiber-modified concrete, the researchers tested analyze the fiber concrete composites under the influence of different external conditions (temperature [11,12], external cyclic loading [13,14], humidity, etc.) and established the analytical model of the stress–strain [15], dynamic intrinsic damage law [16], and intrinsic model [17–19] for basalt fiber concrete composites.

Since basalt fibers are effective for concrete, researchers have tried to apply basalt fibers to the soil. Scholars have undertaken relatively less research on enhancing soils with basalt fibers. Based on the results of unconsolidated undrained triaxial tests, the literature [20] concludes that the undrained shear strength of the pulverized soils increases with the addition of basalt fibers and that the fiber length has a substantial effect on the undrained shear strength of pulverized soils. The authors of [21] demonstrate, through experimental studies, that the tensile strength of soils incorporated with basalt fibers under load is enhanced because the form of stresses to which they are subjected is changed and the composite structure is optimized. The three-dimensional structure generated between the fibers prevents the soil from deforming, promotes a section between soil particles and fibers, and enhances the mechanical properties of the soil [22]. The authors of [23] show that dry density influences the effect of basalt fibers on soil tensile strength. A high dry density encourages contact between soil and fibers, whereas a low dry density allows the soil to be filled with basalt fibers; nonetheless, a low dry density improves the tensile strength of soil more. The authors of [24] reveal that, when basalt fibers were incorporated into cement-amended soils, the cohesion significantly increased with the number of freeze–thaw cycles under the same conditions. The internal friction angle tended to decrease first and then increase in comparison to the cement-only amended soils. The temperature has a non-negligible influence on the mechanical properties of clay, and can even change its properties [25–27]. The authors of [28] show that the triaxial compression tests and scanning electron microscopy experiments were used to investigate the static mechanical properties of pulverized coal modified with basalt fiber and basalt powder under freeze–thaw cycles. The results revealed that different numbers of freeze–thaw cycles had different effects on the soil. The development of a simplified ontogenetic model helped in describing the coupled chemical mechanics of saturated soils [29]. The numerical analysis model of basalt fiber-reinforced soil was developed utilizing the tendon-soil separation method described in the literature [30]. According to the results of the numerical simulations, incorporating basalt fiber can greatly increase the unconfined compressive strength of the soil and decrease the amount of lateral bulge deformation of the soil [31]. The study determined that the damage pattern of the triaxial specimens with high fiber contents changed from shear bulge damage to shear zone damage with the number of dry and wet cycles increased. The continuous damage theory proposed a statistical damage model of fiber-reinforced loess considering dry and wet and load damage [32]. The study examined the influence of freeze–thaw cycles on the unconfined compressive strength (UCS) and P-wave velocity ($V_p$) of lime-stabilized basalt fiber-reinforced loess. The fitting findings indicate that $V_p$ can be used to estimate the UCS after freeze–thaw damage. Not only are the research results applicable to the utilization of basalt fibers in geotechnical engineering, but they also provide a reference for the nondestructive assessment of the strength of loess after freeze–thaw cycles. The authors of [33] illustrate triaxial tests that were used to determine the strong foundation of

basalt short-cut fiber (BCF)-reinforced red clay. The results indicated that the reinforcing impact of BCF on red clay was mostly due to its cohesion improvement. In [34], it was discovered that the strength of the cement soil mixed with fiber was greater than that of the cement soil without fibers. This paper also exhibited good frost resistance via the unconfined compressive strength test and freeze–thaw cycle test, proving that the fibers could enhance the cement soil's properties against freeze–thaw cycles. Free shrinkage and dry cracking were captured using a soil-bound ring test method incorporating digital image correlation, and it was found that reinforcing bentonite with 1.0% using weight-weight fiber was effective in reducing crack extension and separation [35,36]. In [37], by adding different proportions of fly ash, sand, and basalt fibers to expansive soils, it was found after a series of experiments that the entrapment between fibers and soil particles significantly increased the strength of stabilized soils. In addition, due to the impact of sand and fibers during mixing or compression, the surface of fibers became more rugged and the friction between fibers and soil particles increased directly, which had a favorable effect on the stability of expanded soil. In [38], a series of unconfined compressive strength tests were conducted on basalt fiber-reinforced clay soils with the basalt fiber content and length as variables. The experimental results indicate that the basalt fibers can successfully improve the UCS of clay, and the results also indicate that the basalt fiber-reinforced clay has "post-strengthening" properties. Regarding the reinforcement mechanism, a network model of fiber-soil columns was proposed. On the based model and SEM images, the impacts of the fiber content and length are associated with the variation of the fiber-soil column and the formation of an effective fiber-soil network. The authors of [39] utilized a digital image-based triaxial shear test system to explore the effect of dry and wet action of basalt fiber-reinforced loess on the mechanical behavior and damage mode. The results demonstrated that the shear strength of the fiber-reinforced loess reduced during dry and wet cycles, while the fiber content exhibited an inverted U-shaped variation, implying the effectiveness of fibers in inhibiting the sprouting and extension of apparent cracks. The failure mode changed from brittle shear failure to an overall bulge after fiber addition, and the fibers inhibited the formation and propagation of microcracks, with most damage happening during the early stages of the wet and dry cycles.

In summary, the researchers have achieved good improvement results by blending basalt fibers into chalk, loess, swelling soil, cement-improved soil, and other types of soils. However, for soils with special characteristics such as red clay soils, the influence of different factors on the properties of red clay soils improved by basalt fiber is less studied than when mixed with basalt fiber. Therefore, this paper will investigate the strength variation law of fiber-modified red clay with different doping at different dry densities using a direct shear test and unconfined compressive strength test.

## 2. Experimental Material and Methods

### 2.1. Material Selection

The red clay utilized in the experiment was obtained from an industrial site area in Yanshan Town, Guilin City. The soil samples were dried and broken up, sieved, and evaluated for their fundamental physical properties. The data in Table 1 below were obtained.

**Table 1.** Basic physical properties of red clay.

| Liquid Limit | Plastic Limit | Plasticity Index | Optimum Moisture Content | Maximum Dry Density | Specific Gravity |
|---|---|---|---|---|---|
| $W_L$/% | $W_P$/% | $I_P$ | $\omega$/% | $\rho_d/(g \times cm^{-3})$ | $G_S$ |
| 54.63 | 34.82 | 19.81 | 30 | 1.54 | 2.72 |

The selected basalt fibers are from Shanghai Chenqi Chemical Co., Ltd. (Shanghai, China). In determining the fiber length and incorporation rate based on the current scientific research progress on basalt fiber-improved soil, the optimal fiber length is about 6 mm,

and the fiber content is set to 0.2%, 0.3%, and 0.4%, respectively. The fundamental physical indexes are listed in Table 2.

**Table 2.** Basic properties of basalt fibers.

| Length mm | Density ($g \times cm^{-3}$) | Diameter μm | Melting Point | Tensile Strength | Tensile Modulus of Elasticity |
|---|---|---|---|---|---|
| 6 | 2.699 | 17.4 | 1450 °C | >2000 Mpa | >85 Gpa |

*2.2. Specimen Preparation*

2.2.1. Preparation of Direct Shear Test Specimens

The soil samples were dried using the air-drying method; the air-dried red clay was crushed by physical means, and the soil particles with a particle size of less than 2 mm were screened for testing according to the sieving method (as specified in the Geotechnical Test Methods Standard (GB/T50123-2019) [40]). The dry density gradients of 1.35 g/cm$^3$, 1.4 g/cm$^3$, and 1.45 g/cm$^3$ were set according to the measured optimum moisture content and maximum dry density of the plain red clay, and the basalt fibers and air-dried red clay sieved to 2 mm were weighed according to the blending rates of 0.2%, 0.3%, and 0.4%.

When too much basalt fiber is added to the soil at one time, it easily clumps together; so, firstly, the basalt fiber is separated into filaments and the required amount of basalt fiber is divided into portions and added one by one. Spray water once at a time at an optimum moisture content of 30%, manually mix utilizing a stirrer until homogeneous, place the completed soil sample in a sealed bag, and leave it airtight and moist for 24 h so that the soil particles are evenly saturated with water. Next, retest the moisture content to determine that the error is within ±0.5% of the target range to start sample preparation. According to the geotechnical test procedure, the diameter of the pressing ring knife is 61.8 mm, the height is 20 mm, the sample saturation method is saturated according to the extraction saturation method in the "Geotechnical Test Method Standard" [40], and the soil sample is obtained by using the mold release device to break off the ring knife after successful sample preparation. The specimens produced by the hydrostatic method are shown in Figure 1.

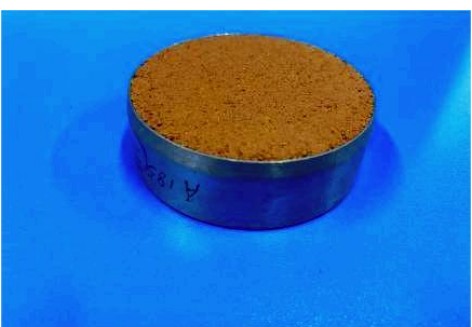

**Figure 1.** Direct shear test specimens.

2.2.2. Preparation of Specimens for the Unconfined Compressive Strength Test

The standard unconfined compressive strength test specimens with a diameter of 39.1 mm and a height of 80 mm were prepared using the layered compaction method [40], with controlled dry densities of 1.35 g/cm$^3$, 1.40 g/cm$^3$, and 1.45 g/cm$^3$, moisture content of 30% and basalt fiber content of 0%, 0.2%, 0.3%, and 0.4%, respectively. The mixing process of the fiber is the same as that of the direct shear test specimen preparation, and the number of specimens to be prepared is 48. There are no lateral limit compressive strength specimens, as shown in Figure 2.

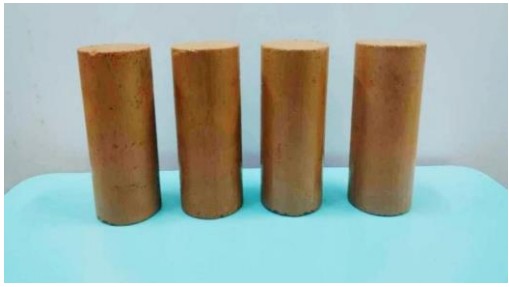

**Figure 2.** Unconfined compressive strength specimens.

## 3. Mechanical Properties of Basalt Fiber-Modified Red Clay

### 3.1. Direct Shear Test

The instrument used in this experiment was the ZJ-type strain-controlled quadruple straight shear instrument of the Nanjing Soil Instrument Factory (Nanjing, China) (Figure 3). Four kinds of vertical pressures were applied to the straight shear specimen of 100 kPa, 200 kPa, 300 kPa, 400 kPa; the control shear speed was 0.8 mm/min; and the shear displacement was 6 mm under the condition of the fast shear experiment. The calculation formula of shear strength is as follows:

$$\tau_f = c + \sigma tan\varphi \tag{1}$$

where $\tau_f$ is the shear strength of the soil (kPa); $c$ is the cohesion of the soil (kPa); $\sigma$ is total normal stress on the shear sliding surface (kPa); and $\varphi$ is the angle of internal friction. The shear strength properties of the red clay are shown in Table 3.

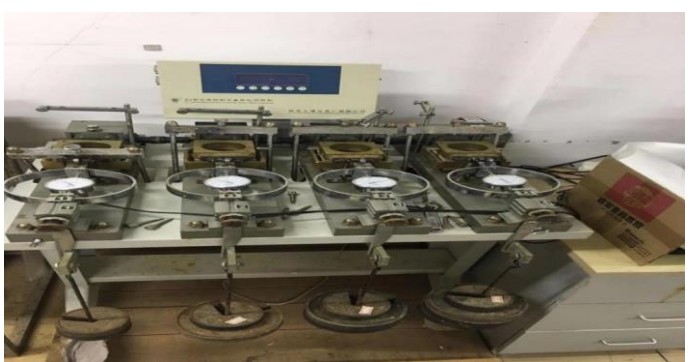

**Figure 3.** The direct shear test device.

**Table 3.** Shear strength properties of red clay.

| Soil Sample Status | | Shear Strength Corresponding to Different Vertical Pressures/kPa | | | | C/kPa | $\varphi$ |
|---|---|---|---|---|---|---|---|
| | | 100 | 200 | 300 | 400 | | |
| Red clay | 1.35 g/cm$^3$ | 43.74 | 85 | 117 | 150 | 7.5 | 19.8 |
| | 1.40 g/cm$^3$ | 47.7 | 87.84 | 121.6 | 154.8 | 14.22 | 19.69 |
| | 1.45 g/cm$^3$ | 50.09 | 90.24 | 126.9 | 156.8 | 16.81 | 19.635 |

Under the same vertical pressure, the shear strength of vegetal red clay grows continuously with the increasing dry density. There is a positive correlation between the shear strength and vertical pressure in the same dry density case.

### 3.2. Effect of Blending Basalt Fibers into Red Clay on Shear Strength

Direct shear tests were used to determine the shear strengths of different basalt admixture contents at varying dry densities under four vertical pressures, as shown in Table 4 below.

**Table 4.** Shear strength of basalt fiber-modified red clay.

| Soil Sample Status | | Shear Strength for Different Vertical Pressures/kPa | | | | *C*/kPa | *φ* |
|---|---|---|---|---|---|---|---|
| | | **100** | **200** | **300** | **400** | | |
| 0.2% basalt fiber-modified red clay | 1.35 g/cm³ | 48.78 | 92 | 133.2 | 156 | 16.78 | 19.942 |
| | 1.40 g/cm³ | 51.49 | 98.25 | 136.9 | 161.5 | 19.87 | 20.238 |
| | 1.45 g/cm³ | 56.78 | 104.3 | 140.2 | 168 | 24.93 | 20.272 |
| 0.3% basalt fiber-modified red clay | 1.35 g/cm³ | 51.22 | 97.76 | 139.88 | 158 | 21.10 | 19.945 |
| | 1.40 g/cm³ | 56 | 100 | 143 | 164.6 | 23.70 | 20.244 |
| | 1.45 g/cm³ | 62 | 101.9 | 146.6 | 172.7 | 26.60 | 20.601 |
| 0.4% basalt fiber-modified red clay | 1.35 g/cm³ | 50.2 | 89.32 | 130.2 | 155 | 17.36 | 19.502 |
| | 1.40 g/cm³ | 52.69 | 90.79 | 135.6 | 160.8 | 17.68 | 20.251 |
| | 1.45 g/cm³ | 58.6 | 94.2 | 138.5 | 169.7 | 20.85 | 20.598 |

1. Ensure that the basalt fiber incorporation rate is constant and study the variation of the effect on its shear strength under different dry density cases.

We have plotted the following four images in Figure 4 based on the data in Table 4.

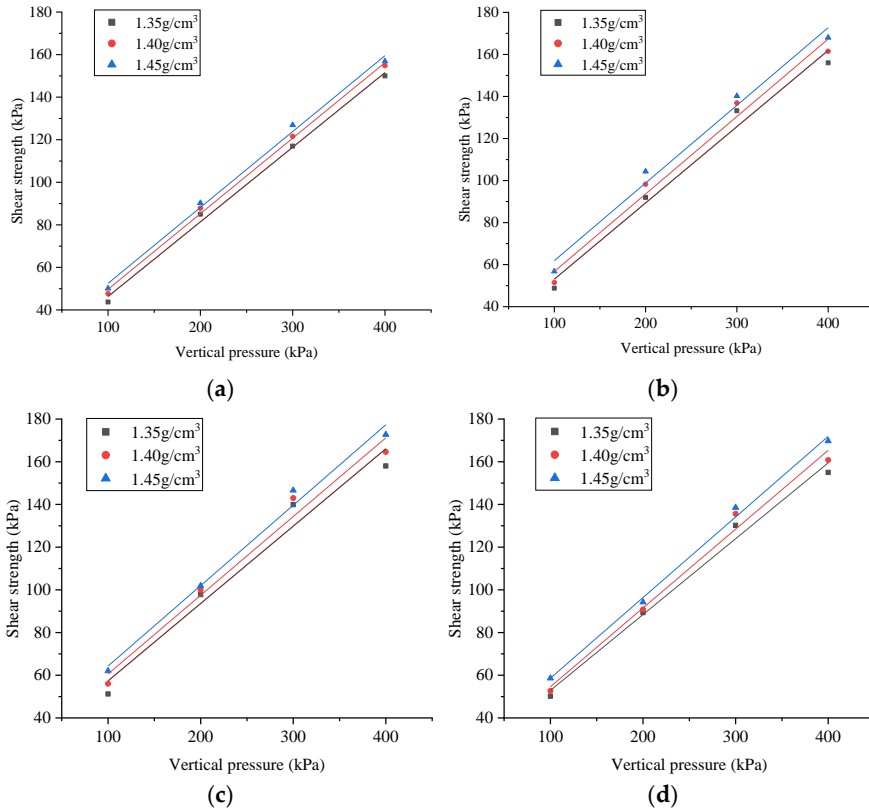

**Figure 4.** Shear strength of basalt fiber-modified red clay under different vertical pressures. (**a**) plain red clay. (**b**) 0.2% fiber-doped red clay. (**c**) 0.3% fiber-doped red clay. (**d**) 0.4% fiber-doped red clay.

When basalt fibers with the same doping content were used for quantitative analysis, it was discovered that the shear strength of the red clay increased with the increase in the dry density. Additionally, the increase in the shear strength from 1.35 g/cm³ to 1.40 g/cm³ dry density after fiber incorporation is less than from 1.40 g/cm³ to 1.45 g/cm³. This is because, under conditions of low dry density, there are large intervals between the red clay soil particles, and a large number of small pores and the shear stress of interaction is small when

subjected to an external force. With an increase in the dry density, the interval pore space reduces, the shear stress increases, and the resultant shear strength improves dramatically. The fiber is not in sufficient contact with the soil at a dry density of 1.40 g/cm$^3$, so the shear strength does not increase much under vertical pressure at this time, but the soil particles are basically in full contact within the soil at from 1.40 g/cm$^3$ to 1.45 g/cm$^3$, so the shear strength naturally increases more.

2.  Keeping the dry density constant, the effect of different fiber incorporation rates on the shear strength was studied.

The three plots in Figure 5 reveal that the shear strength of the red clay with a basalt fiber admixture is greater than the shear strength of the red clay plain clay at various vertical pressures. Additionally, in the same dry density case, the shear strength of the 0.3% basalt fiber admixture is the greatest, while the shear strengths of the 0.2% and 0.4% basalt fiber admixtures are close in size, with the shear strength of the 0.2% basalt fiber admixture being somewhat greater. Because the interaction between the fibers and the soil is relatively adequate in the case of the optimal basalt fiber admixture, the soil's overall shear strength rises the most. For the 0.4% basalt doping, the doping rate is too high, soil fibers overlap, and the interaction effect between the soil particles and basalt fibers diminishes, so the overall shear strength of the soil body grows the least.

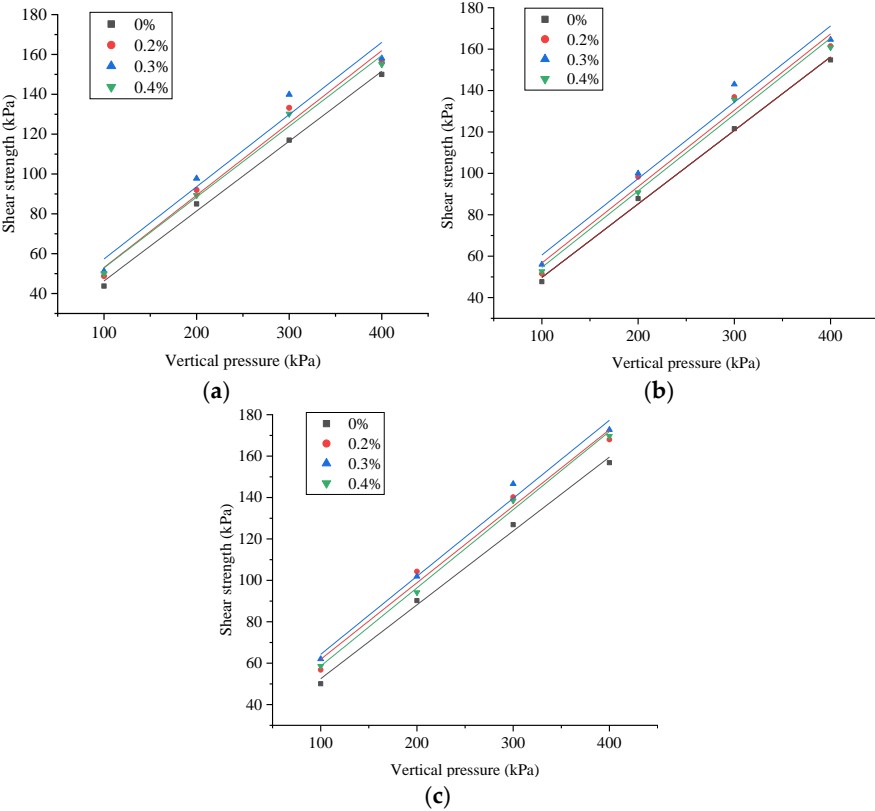

**Figure 5.** Shear strength of red clay-modified by basalt fibers with different incorporation rates. (**a**) Dry density of 1.35. (**b**) Dry density of 1.40. (**c**) Dry density of 1.45.

3.  Effect of fiber incorporation rate on the cohesion and internal friction angle of basalt fiber-modified red clay.

It can be found from Figure 6 that the cohesive force of red clay increases and then decreases with the rise of the incorporation rate after the incorporation of basalt fiber with the same dry density and increases with the rise of the dry density with the same incorporation rate. At a dry density of 1.35 g/cm$^3$, the increase in the cohesion of the red clay with 0.2%, 0.3%, and 0.4% basalt fiber was 123.7%, 181.3%, and 131.4%, respectively; at

a dry density of 1.40 g/cm³, the increase in the cohesion of the red clay with 0.2%, 0.3%, and 0.4% basalt fiber was 39.7%, 66.7%, and 24.3%, respectively; and at a dry density of 1.45 g/cm³, the increase in the cohesion of the red clay with 0.2%, 0.3%, and 0.4% basalt fiber was 39.7%, 66.7%, and 24.3%, respectively. At a dry density of 1.45 g/cm³, the increase in the cohesion of red clay with 0.2%, 0.3%, and 0.4% basalt fiber was 48.3%, 58.2%, and 24.1%, respectively. The increase in the cohesion of basalt fiber-modified red clay compared with that of the plain red clay for the same dry density increased first and then decreased.

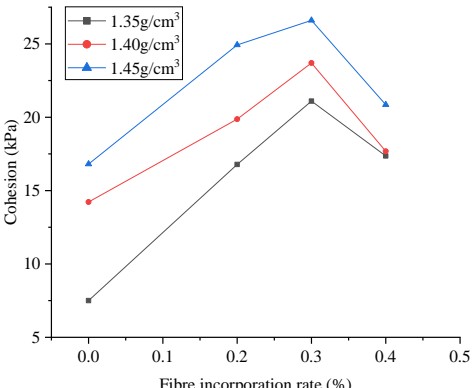

**Figure 6.** The cohesion of different incorporation rates.

After a small amount of basalt fibers are mixed in, the fibers are in full contact with the soil and the fine soil particles are adsorbed on the fibers and agglomerated into fibrous soil columns. After receiving the external shear forces, the friction between the soil and the soil and the friction between the fibers and the soil counteracts the external shear forces, and the shear strength of the soil is increased. When 0.3% is the critical value of the soil mixed with the fibers, at this time, the soil particles have been in full contact with a single fiber, so the friction between soil particles and fibers is increasing, the soil is full of fiber soil columns, and the cohesion of the soil reaches its maximum. Beyond the critical value, continue to add basalt fibers, and the fibers will begin to stagger and overlap to form a network, i.e., a fiber-soil network, but cause the soil to be incapable of being dense and the interaction between fibers to be offset. At the same time, the interaction between the fibers counteracts the friction between the fibers and the soil, causing the cohesion to decrease.

It can be found from Figure 7 that the internal friction angle increases and then decreases with the increasing incorporation rate at a dry density of 1.35 g/cm³ when basalt fibers are incorporated into the red clay. At dry densities of 1.40 g/cm³ and 1.45 g/cm³, the internal friction angle increases with the incorporation rate and then remains stable with small fluctuations.

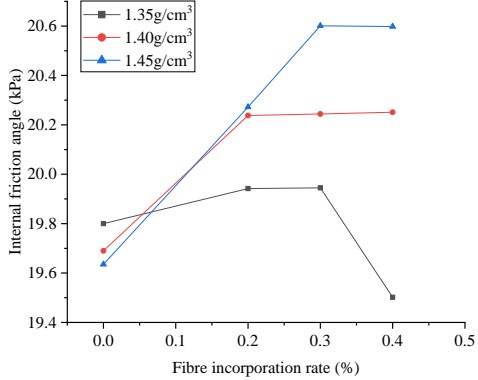

**Figure 7.** Internal friction angle of different incorporation rates.

The angle of internal friction is greatest at a dry density of 1.45 g/cm$^3$ with a basalt fiber incorporation rate of 0.3%. This indicates that basalt fibers are incorporated into the soil precisely by forming a sinew-soil interface, filling the pores, bringing the soil particles into full contact with the fibers, strengthening the friction between the soil particles and between the fibers and the soil particles, increasing the internal friction angle of the soil, and ultimately increasing the shear strength and cohesion of the soil. At low dry densities, the soil particles and some fibers do not have sufficient contact with each other, which can damage the friction between the soil particles and affect the internal friction angle, resulting in a lower friction angle than that of plain soil at low dry densities.

### 3.3. Unconfined Compressive Strength Test

The determination of the unconfined compressive strength of the red clay mixed with basalt fibers by strain-controlled unconfined compression apparatus. The loading procedure for the unconfined compressive strength test is shown in Figure 8.

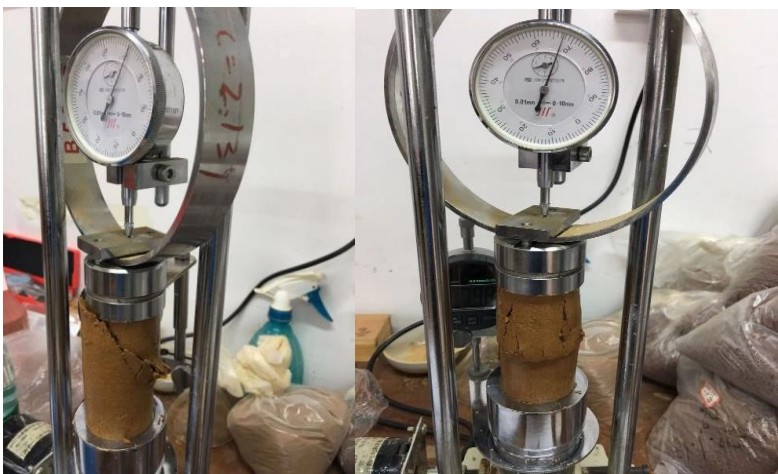

**Figure 8.** Loading procedure for unconfined compressive strength tests.

The unconfined compressive strength is one of the most important mechanical property factors. The calculation formula is as follows:

$$\sigma = \frac{C * R}{A_a} \times 10 \tag{2}$$

where $\sigma$—axial stress (kPa); $C$—force gauge rate coefficient (N/0.01 mm); $R$—force gauge reading (0.01 mm); and $A_a$—the area of the cross-section of the specimen in shear (cm$^2$).

Table 5 displays the results of the unconfined compressive strength tests conducted on the plain red clay to determine its compressive strength at various dry densities.

**Table 5.** Red clay unconfined compressive strength properties.

| Soil Sample | Dry Density | Reinforcement Rate% | Peak Intensity $\sigma$/kPa | Residual Strength $\sigma$/kPa | Peak Strain $\varepsilon$/% |
|---|---|---|---|---|---|
| Vegetable red clay | 1.35 g/cm$^3$ | 0% | 89.08 | 48.90 | 2.5% |
| | 1.40 g/cm$^3$ | 0% | 102.52 | 74.72 | 2.5% |
| | 1.45 g/cm$^3$ | 0% | 121.91 | 91.43 | 2.5% |

### 3.4. Effect of Basalt Fibers on Unconfined Compressive Stress and Strain Curves of Red Clay

1.　Variation of stress–strain curves for different dry densities at the same admixture rate.

In Figure 9, With the change of the dry density of red clay with different doping rates, each axial stress–strain graph was observed, and it was found that there was no obvious

linear change trend pattern for each curve. In proportion to the increase in the axial strain, the stresses exhibited a quick increase before a strain of approximately 4%, followed by a gradual decline. This stress–strain graph resembles the strain-softening graph. With an increase in the soil's dry density, the ductility increases.

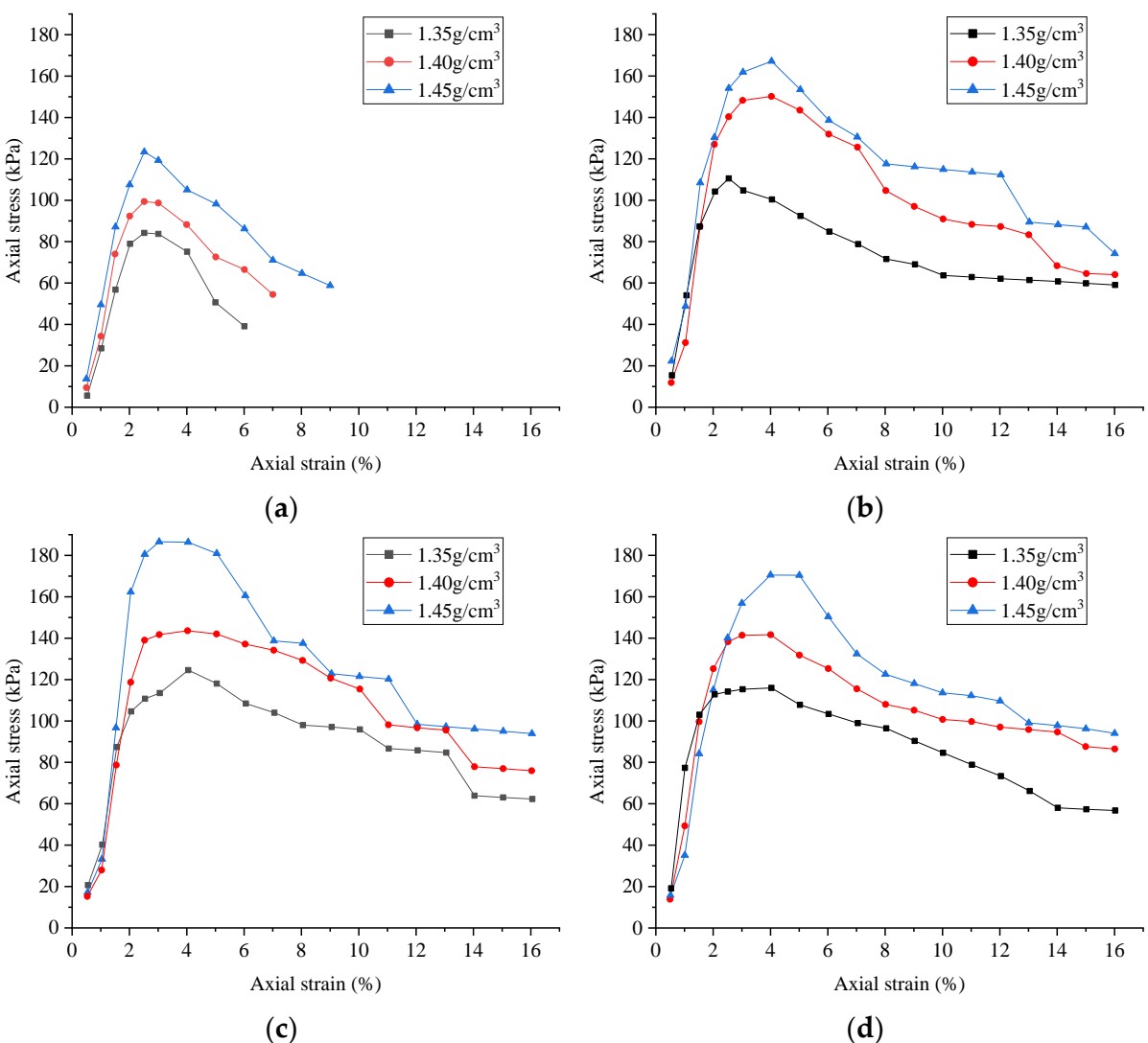

**Figure 9.** Stress–strain curve of basalt fiber-modified red clay with different dry densities. (**a**) Plain clay. (**b**) 0.2% fiber-doped red clay. (**c**) 0.3% fiber-doped red clay. (**d**) 0.4% fiber-doped red clay.

2.  The stress–strain curve changes under different fiber incorporation contents under the same dry density conditions.

The pattern of change is found in Figure 10. In the case of uniform dry density, the stress–strain curves of red clay all grow to a peak first and then decrease smoothly, whereas the curve with fiber incorporation resembles a softening-type curve and has a slow decay rate. This is because the fibers are interconnected with the soil, and, even after the soil is cracked, the fibers will prevent the soil from deforming, thus delaying the loss of strength.

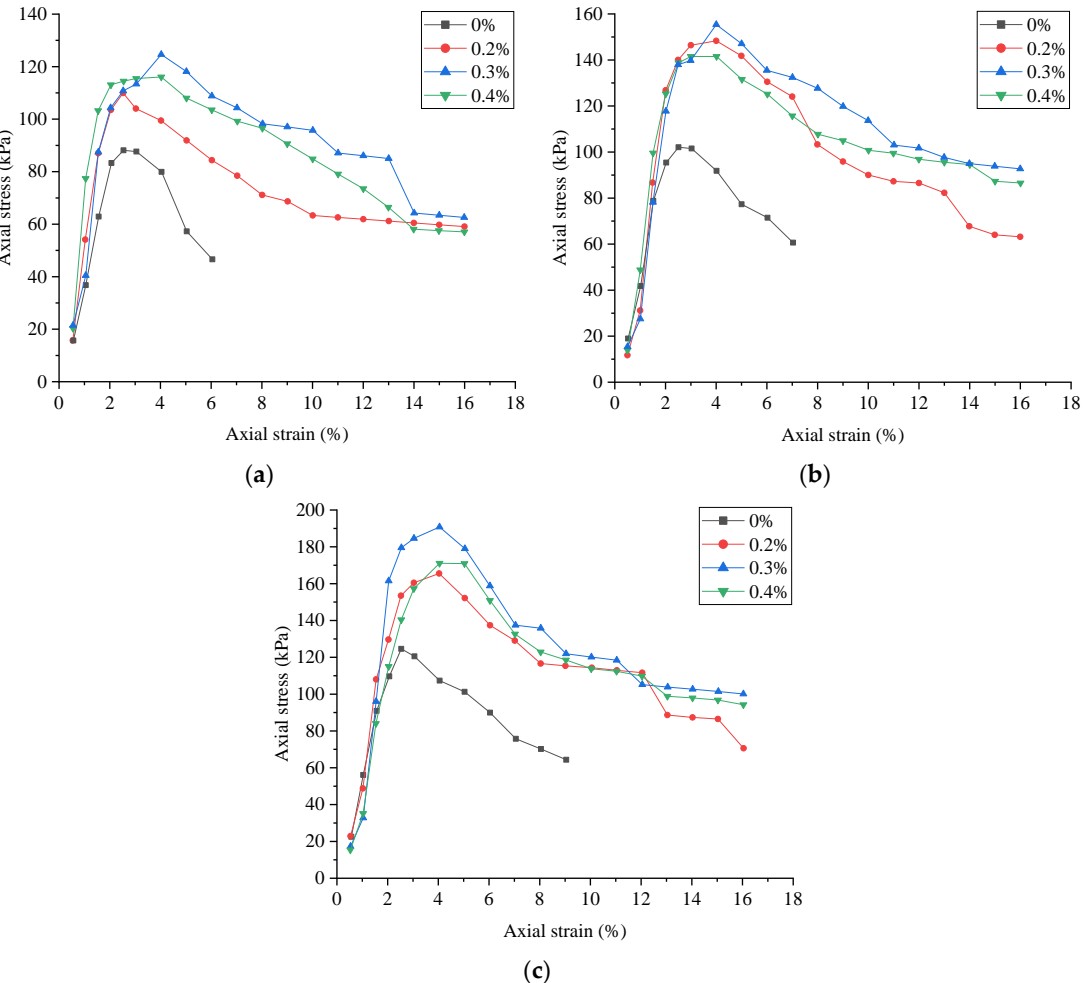

**Figure 10.** Stress–strain curves of basalt fiber-modified red clay with different incorporation rates. (**a**) Dry density of 1.35 g/cm$^3$. (**b**) Dry density of 1.40 g/cm$^3$. (**c**) Dry density of 1.45 g/cm$^3$.

*3.5. Analysis of the Effect Law of Blending Basalt Fibers into Red Clay on the Variation of Unconfined Compressive Strength and Residual Strength*

The unconfined compressive strength test was used to assess the peak and residual strengths of varied basalt admixture contents under vertical pressure at various dry densities, as shown in Table 6 below.

**Table 6.** Variation of unconfined compressive strength of basalt fiber-modified soil.

| Soil Sample | Dry Density | Reinforcement Rate% | Peak Intensity σ/kPa | Residual Strength σ/kPa |
|---|---|---|---|---|
| 0.2% basalt fiber-modified red clay | 1.35 g/cm$^3$ | 0.2% | 109.50 | 87.26 |
| | 1.40 g/cm$^3$ | 0.2% | 137.98 | 111.78 |
| | 1.45 g/cm$^3$ | 0.2% | 165.93 | 121.91 |
| 0.3% basalt fiber-modified red clay | 1.35 g/cm$^3$ | 0.3% | 124.90 | 102.66 |
| | 1.40 g/cm$^3$ | 0.3% | 149.00 | 133.76 |
| | 1.45 g/cm$^3$ | 0.3% | 191.33 | 145.97 |
| 0.4% basalt fiber-modified red clay | 1.35 g/cm$^3$ | 0.4% | 98.81 | 92.53 |
| | 1.40 g/cm$^3$ | 0.4% | 142.01 | 112.92 |
| | 1.45 g/cm$^3$ | 0.4% | 171.01 | 123.60 |

### 3.5.1. Effect of Fiber Incorporation Rate on Unconfined Compressive Strength

The effect of different fiber incorporation rates on peak and residual strengths under the same dry density conditions.

The following findings can be analyzed based on what can be seen in Figures 11 and 12. At a dry density of 1.35 g/cm$^3$, the increase in the unconfined compressive strength of the red clay with 0.2%, 0.3%, and 0.4% basalt fiber was 22.93%, 40.00%, and 10.92%, respectively. At a dry density of 1.40 g/cm$^3$, the increase in the unconfined compressive strength of red clay with 0.2%, 0.3%, and 0.4% basalt fiber was 34.58%, 45.0%, and 38.51%, respectively, and, at a dry density of 1.45 g/cm$^3$, the increase in the unconfined compressive strength of red clay with 0.2%, 0.3%, and 0.4% basalt fiber was 36.11%, 55.00%, and 40.28%, respectively. The peak compressive strength and residual strength of the red clay grew continuously when the incorporation content was less than 0.3% and the same dry density but then began to progressively drop. The soil's peak and residual strengths increased as its dry density increased. This is because the basalt fibers incorporated into the soil increase the soil's toughness, resulting in ductile damage as the final form of damage for the modified soil. Through the fold line in the soil, it is evident that the soil has the maximum peak strength and residual strength with 0.3% of the basalt fiber admixture and 1.45 dry soil density.

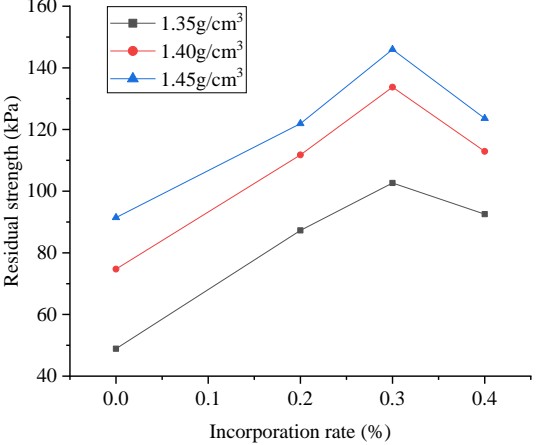

**Figure 11.** The peak strength of different incorporation.

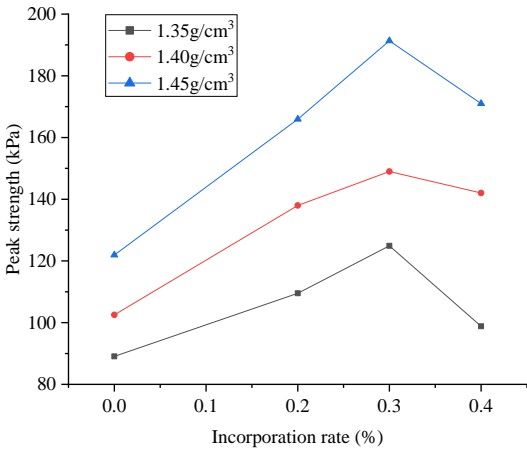

**Figure 12.** Residual strength of different incorporation rates.

### 3.5.2. Effect of Dry Density on Unconfined Compressive Strength

The effect of different dry densities on the peak and residual strengths at the same admixture rate conditions.

The pattern of change can be found in Figures 13 and 14. The increase in the dry density is greater at from 1.35 g/cm³ to 1.40 g/cm³ than at from 1.40 g/cm³ to 1.45 g/cm³ for plain soils and 0.3% for basalt fiber-modified red clay, while the opposite is true for the 0.2% and 0.4% for basalt fiber-modified red clay. The reason for this is that the increase in the dry density in the case of plain clay is by its very nature a reduction in the voids between the soil particles, causing the soil to be more homogeneous. The peak unconfined compressive strength and residual strength of the soil at the same admixture content continuously increase with the increase in the dry density, and when the admixture content is 0.3%, the increment of the peak strength is greater with the increase in the dry density. As a result of the optimal mixing, the soil has attained a relatively tight interior with a low dry density, with full contact between the soil particles and fibers, and substantial variations in the peak strength. The other additives gradually touch the fibers as the dry density increases and resist compression. In contrast, the residual strength is proportional to the quantity of doping amounts; since the soil is fully compacted at the time of destruction, there is no increase in the strength gain even if the dry density increases.

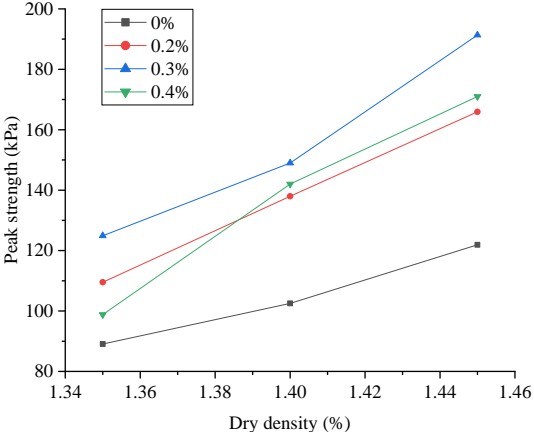

**Figure 13.** The peak strength of different dry density.

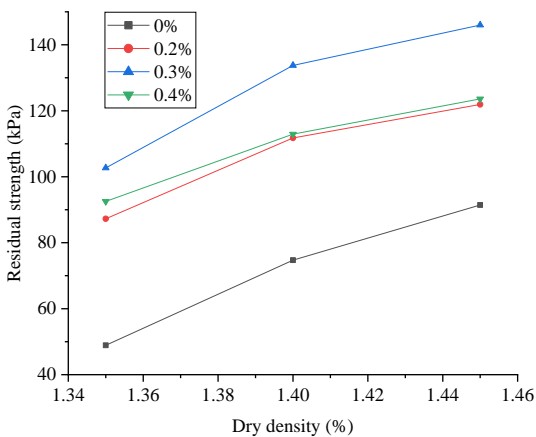

**Figure 14.** Residual strength of different dry density.

## 4. Microstructure Analysis

Figure 15 depicts the microscopic electron microscopy images of the red clay soil scanned at two different magnifications, revealing that the soil body is formed of various sizes and shapes soil particles. Due to its rough surface, there are more pores and fewer tight connections between the particles. When the cementing material and water film between the soil particles are destroyed, the internal microstructure changes and, eventually, the overall destruction occurs.

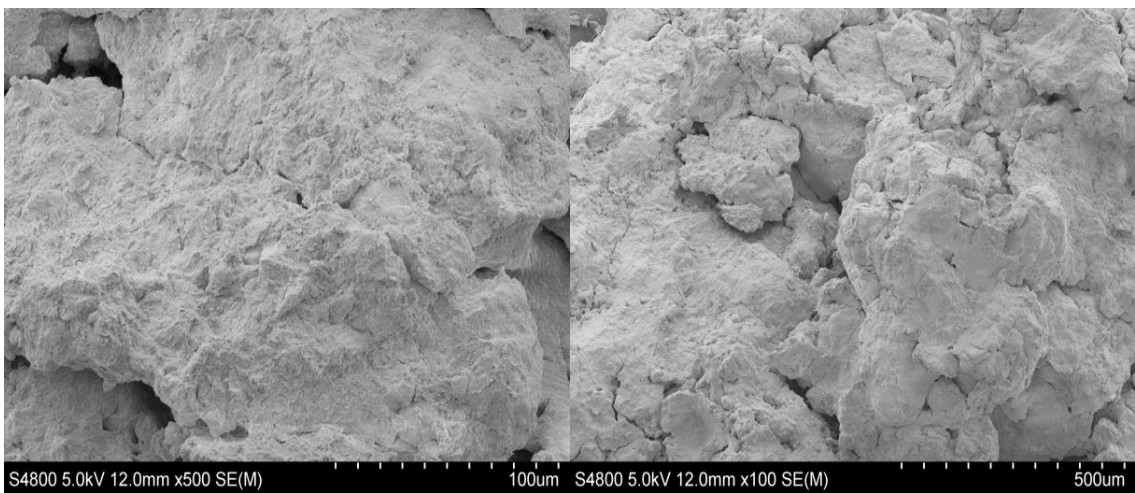

**Figure 15.** Microstructure of undoped fiber red clay.

Figure 17 demonstrates that when the red clay is mixed with only 0.2% of basalt fibers, the fibers are scattered in the soil particles alone. Basically, there is only a single interaction between some soil particles and fibers, and the mechanical properties of the soil body are diminished. When the incorporation rate reaches the optimal rate of 0.3%, the fibers cross each other in the soil particles, establishing an effective force network and filling the pores between the soil particles. The cohesion and friction are greatly enhanced, and the soil's mechanical characteristics are fully enhanced. However, when the incorporation of basalt fibers is increased to 0.4%, the basalt fibers are stacked and do not form an effective connection with the soil, nor do they adequately fill the pores. The mechanical characteristics have deteriorated.

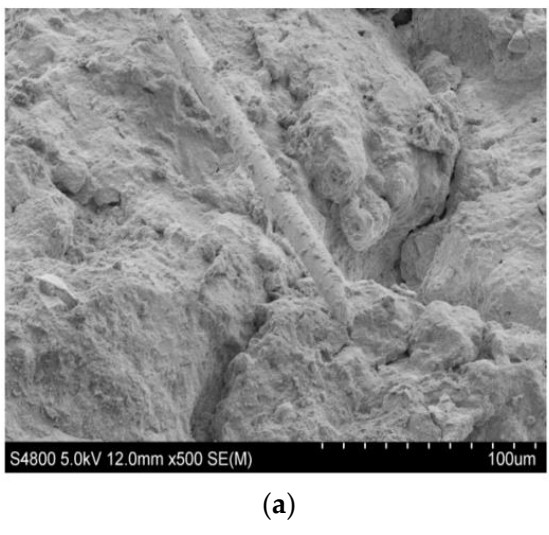

(**a**)

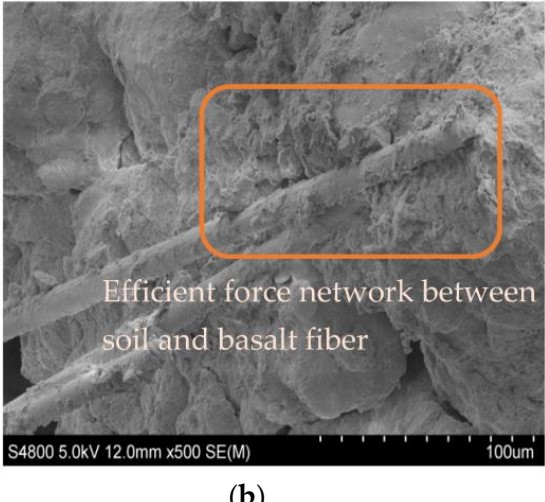

(**b**)

**Figure 16.** *Cont.*

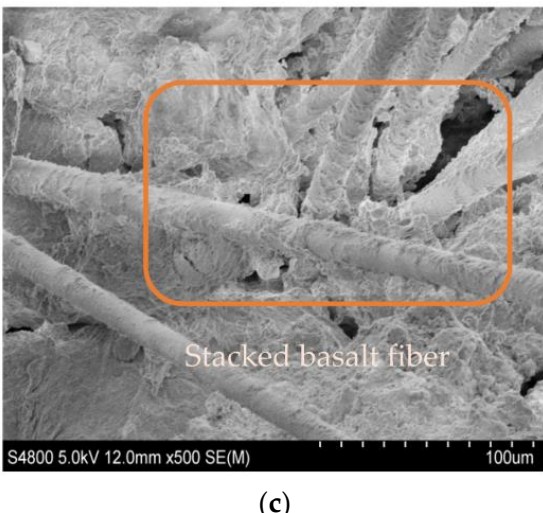

(**c**)

**Figure 17.** SEM diagram of red clay with different basalt fiber doping. (**a**) SEM image of red clay mixed with 0.2% basalt fiber. (**b**) SEM image of red clay mixed with 0.3% basalt fiber. (**c**) SEM image of red clay mixed with 0.4% basalt fiber.

From a microscopic perspective, the mechanism of the effect of the incorporation rate on the mechanical properties of red clay is analyzed. When the basalt fiber content is low and the fiber spacing is large, it is difficult for the fiber-soil body to intermingle with each other, preventing the fiber and soil from forming a fiber-linked soil network. When the soil is subjected to external forces, it is mainly transferred and carried through the forces between the dispersed fiber-soil body and the soil particles themselves. When the basalt fiber content grows gradually, the fiber spacing decreases, allowing adjoining fiber-soil bodies to readily intersect and form a fiber-linked soil network. So, at this time, the soil is subjected to external forces that are primarily transferred and withstood by the forces between the fiber-bonded soil network and the soil particles. The test findings indicate that the reinforcement effect is not good when the basalt fiber content is high. Since the interaction between the fibers and the soil gradually increases with the increasing basalt fiber content and the frictional resistance between the fibers is significantly lower than the frictional force at the interface of the fibers and the soil particles, it is difficult to prevent the overall deformation of the soil from increasing by converting the external force into the internal force of the fibers in the form of the contact surface force. Due to electrostatic interactions, when the basalt fiber is relatively high, a large number of fiber filaments will accumulate within the soil sample, causing it to be difficult to evenly distribute the fibers. The weak zones of stress tend to form, which is not conducive to the transfer of stress. Therefore, the excess fibers will reduce the effect of fiber reinforcement and the mechanical properties of the soil will be reduced.

## 5. Conclusions

The conclusions are as follows:

1. Compared with the plain red clay, the red clay mixed with basalt fibers exhibited a considerable improvement in the compressive strength and shear strength, which peaked at the optimal admixture content of 0.3%.
2. At the optimum incorporation level, the basalt fibers fill the pores in the soil and the cohesion and internal friction angle of the soil are maximized.
3. After compression, it was found that the stress–strain curve of the soil containing basalt fibers remained unchanged, but the strength change on both sides of the optimal incorporation content increased and then decreased. The strength increased with the increase in the dry density for the same admixture content, but the magnitude was small.

4. After the incorporation of basalt fiber, the initial form of damage was transformed into ductile damage, which can maintain high residual strength at the end.

5. Through microstructure tests, the fiber-soil network produced by the fibers and the soil can most effectively function only at the optimal fiber incorporation amount, which means the mechanical properties of the improved soil can be enhanced to the greatest extent.

**Author Contributions:** Conceptualization, Y.S.; validation, S.D. (Song Ding). and K.X.; data curation, S.D. (Shuaishuai Dong). and Y.G.; writing—original draft preparation, Y.G.; writing—review and editing, R.Y. and F.L.; supervision, Y.S.; project administration, R.Y. and F.L. All authors have read and agreed to the published version of the manuscript.

**Funding:** (1) Guangxi Innovation-Driven Development Special Project (Guik AA20161004-1). (2) National Key Research and Development Program Project (2019YFC0507502). (3) National Natural Science Foundation of China (41967037). (4) National Natural Science Foundation of China (42262030).

**Institutional Review Board Statement:** Not applicable.

**Informed Consent Statement:** Not applicable.

**Data Availability Statement:** Not applicable.

**Conflicts of Interest:** The authors declare no conflict of interest.

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
