# Peer review of "Study on Mechanical Properties and Microstructure of Basalt Fiber-Modified Red Clay"

_sustainability, doi:10.3390/su15054411_

Round 1

Reviewer 1 Report

Currently there are uncertainties in the manuscript that need to be clarified. English writing is often confusing and not sufficiently clear. A revision is needed, by someone who is in the field, in order to get a better understanding of the authors ideas.Some additional suggestions are provided below:

1 - The reasoning behind selecting basalt fibers to strengthen clay soil is unclear. The examples shown in the beginning of the introduction are concrete. How do concrete compares to soil in some form, which makes its application to soil as promising? Aren't basalt fibers too sophisticated (they require a lot of energy for production)?

2 - The claim that basalt fibers are eco-friendly - line 33, should be grounded on evidence, with perhaps a LCA or som other environmental impact estimation when compared to alternatives;

3 - How do 'microorganisms decompose the modified soil' - line 36? Do they decompose basalt fibers?

4 - The contribution of the manuscript to cover an existing knowledge gap in the literature should be clarified, in the end of the introduction.

5 - In Fig 3 it would be better to always adotp the same scale for x and y axis, in order to compare different graphs.

6 - The mixing procedure is very important to guarantee a good uniformity. It would be important to include some images and detail further the process.

7 - In order to better understand how the developed mixtures meet certain application requirements, it would be important to include some examples of applications for this technology, as well as requirements in terms of mechanical behaviour. At some point the authors mention fiber-soil columns (line 124), can you further explain the requirements in terms of material properties of this application?

8 - how do we discuss the results obtained and compare them with expectable requirements of real applications?

Reviewer 3 Report

Please see the file in the attachment.

Reviewer 4 Report

Shear strengths of red clay improved by basalt fiber were examined using direct shear and unconfined compression tests. As a result, optimal content of fiver is shown in the manuscript. Description for specimen conditions is insufficient. Moreover, it is difficult to catch the exact meaning because many ambiguous words are used. Therefore, consistent wording is necessary. Following items are specific comments to revise the manuscript.

1. Page 3, line 137 and others: What is "lateral limitless compressive test"? Is that different from unconfined compression test or uniaxial compression test? If those are the same, use the common terminology. If those are different, explain the difference clearly.

2. Page 4, line 153 and 160: "Preparation of direct shear test specimens." is not sentence but something like title. "Preparation of specimens for unconfined compressive strength tests." is also the same. Please carefully check in other parts because there are sentences of the same kind.

3. Page 4, "2.2. Specimen preparation": Please describe specific values of dry density and moisture content for direct shear test specimen. As additional information, degree of saturation and consolidation behaviour before shearing are also important.

4. Table 4: Please describe cohesion and internal friction angle as the same as Table 3.

5. Figures 1 and 2: Please add the unit for dry density.

6. Page 5, line 195 to 197: "friction force" and "friction" are ambiguous words to understand the meanings. The reviewer thinks shear stress can be used instead of those words.

7. Figures 3 to 8: "KPa" should be "kPa".

8. Figure 6: Please reconsider the title referring to Fig. 5.

9. Figures 10(b) and 10(c): It is difficult to see effective force network (Fig. 10(b)) and no effective connection (Fig. 10(c)). Please improve figures by indicating effective connection directly in the figures.

Round 2

Reviewer 1 Report

The authors have addressed most of the comments by the reviewer. English writing needs to be revised, including the new text.

Author Response

Thanks to the reviewers, we adjusted writing issues in the new text and the original text, checked for grammatical problems, uniformly rewritten words that were inconsistent with the new text and the original text, and corrected inappropriate or incorrect words. corrected.

Reviewer 3 Report

No further commens

Author Response

(The authors gave the same response as above.)

Reviewer 4 Report

The reviewer has checked the authors' response and revised manuscript again. The manuscript was revised according to the reviewer's comments. However, the reviewer has found a few minor items that additional revision is needed. Please confirm the following items.

1. Related to the comment No. 1: "lateral limitless test" has still remained in Page 3 line 103.

2. Related to the comment No. 9: Explanatory text was added in the Fig. 16. However, the words used in the figure are different from that used in the main test. Consistent wording is requested.
